# Pediatric Wernicke Encephalopathy: A Systematic Review

**DOI:** 10.3390/pediatric17010015

**Published:** 2025-01-30

**Authors:** Erik Oudman, Jan W. Wijnia, Janice R. Bidesie, Mirjam J. van Dam, Misha J. Oey, Sterre Smits, Maaike van Dorp, Albert Postma

**Affiliations:** 1Experimental Psychology, Helmholtz Institute, Utrecht University, 3584 CS Utrecht, The Netherlands; j.bidesie@leliezorggroep.nl (J.R.B.); mi.vandam@leliezorggroep.nl (M.J.v.D.); m.oey@leliezorggroep.nl (M.J.O.); st.smits@leliezorggroep.nl (S.S.); m.vandorp@leliezorggroep.nl (M.v.D.); a.postma@uu.nl (A.P.); 2Slingedael Korsakoff Center, Lelie Care Group, 3086 EZ Rotterdam, The Netherlands

**Keywords:** Wernicke encephalopathy, Korsakoff syndrome, malnutrition, parenteral nutrition, prevention, vitamin B1, systematic review

## Abstract

Background: Wernicke Encephalopathy (WE), a neurological disorder often linked to alcohol use, can also occur under non-alcoholic conditions, including in pediatric populations. Methods: This systematic review examines 88 pediatric WE cases reported over the past 30 years, encompassing diverse etiologies such as cancer (25 cases), gastrointestinal diseases (19), malnutrition (17), psychiatric disorders (13), obesity surgery (5), renal disease (4), COVID-19 (2), PICU complications (1), hyperemesis gravidarum (1), and a genetic mutation (1). Results: Prodromal symptoms included nausea (60%) and vomiting (55%). In total, 37% of the patients received parenteral nutrition without thiamine before WE diagnosis, often progressing to Wernicke–Korsakoff syndrome (WKS). Key findings revealed the classic triad of WKS, eye movement disorders (80%), mental status changes (75%), and ataxia (63%), with MRI demonstrating high diagnostic sensitivity (85%). Treatment varied widely; higher parenteral thiamine doses correlated with faster recovery and better outcomes, while insufficient dosages led to adverse effects. Full remission was achieved in 61% of cases, with improved outcomes in more recent reports due to refined dosing protocols. Conclusions: These findings underscore the importance of early recognition of nausea and vomiting as predictors of pediatric WE and the critical need to incorporate thiamine in parenteral nutrition for children. Optimal dosing remains vital for recovery, particularly in severe cases.

## 1. Introduction

Wernicke Encephalopathy (WE) is a neurological disorder characterized by thiamine (vitamin B1) deficiency, first delineated by Carl Wernicke in 1881 as a triad of altered mental status, ocular signs, and ataxia, which represents the acute phase of Wernicke–Korsakoff syndrome (WKS). The chronic phase, known as Korsakoff’s syndrome (KS), manifests as an amnesic disorder with confabulations, as described by Sergei Korsakoff in 1887. Traditionally, WKS has been predominantly associated with chronic alcohol use disorder (AUD) due to malnutrition and impaired thiamine absorption. However, it is increasingly recognized that alcohol is not a prerequisite for the development of WKS [1,2,3,4].

Early reports by Wernicke and Korsakoff included several cases of WE and KS unrelated to alcohol consumption, indicating a diverse etiology beyond AUD. Notably, nutritional deficiency emerged as a primary cause, elucidated decades later by Captain De Wardener and Lennox’s observations in prisoners of war [2]. Modern medical contexts such as famine, anorexia nervosa, and certain medical treatments also predispose individuals to WKS due to inadequate thiamine intake. Additionally, conditions involving chronic vomiting or diarrhea, such as hyperemesis gravidarum and inflammatory bowel disease, pose risks for thiamine deficiency and subsequent WKS [1,2,3,4,5,6,7].

PWE was first described by Renato Guerrero in 1949. It presents unique challenges and considerations, such as a presentation without the classic triad, underdiagnosis, and relatively nonspecific symptoms. Infants, children, and teenagers are also susceptible to thiamine deficiency due to factors such as inadequate dietary intake, malabsorption syndromes, chronic illnesses, or parenteral nutrition devoid of thiamine supplementation. Additionally, pregnant teenage mothers diagnosed with hyperemesis gravidarum, inherited metabolic disorders, and pediatric malignancies can predispose children to WE [8,9]. Importantly, the clinical etiologies and manifestations of PWE are expected to diverge from those observed in adults. For instance, previous systematic reviews focusing on non-alcohol-use-disorder cases of Wernicke Encephalopathy have indicated that young age may confer a protective effect against the onset of cognitive sequelae associated with the condition [10,11]. Additionally, genetic factors or developmental conditions could potentially be overrepresented in cases of PWE [4].

Prompt recognition and treatment of PWE are critical to prevent long-term neurological damage. The vulnerability of the developing brain in children demands a high level of clinical awareness, particularly as symptoms can be nonspecific, and the classic triad of symptoms may be absent. Timely diagnosis and intervention can significantly enhance recovery chances and reduce the risk of residual damage, such as that observed in Korsakoff syndrome in adults [1,2,3,4,8]. It is therefore essential for clinicians to remain vigilant about the risks of thiamine deficiency in pediatric patients, enabling optimal care and minimizing potential long-term neurological sequelae [9,11]. Given the vulnerability of developing brains, prompt recognition and treatment of WE in pediatric populations are imperative to prevent long-term neurological sequelae. Pediatric forms of WE warrant special attention and further investigation to optimize clinical management and outcomes. We therefore performed a systematic review to investigate the signs and symptoms of PWE. Our primary goal was to increase clinicians’ awareness and recognition of non-alcoholic pediatric Wernicke Encephalopathy, by describing prodromal signs, symptomatology, method of diagnosis, and outcome of treatment.

## 2. Materials and Methods

### 2.1. Study Design

Our research protocol was registered in PROSPERO (CRD42024549662). There was no funding for this project. We performed a systematic review of the literature. We included systematic reviews reporting on WKS in pediatric non-alcoholic patients. Reports were considered for inclusion if patients were younger than 18 years of age and at least one of the following methods of diagnosing WE was reported and consistent with the findings reported in the case description: Caine’s operational criteria for WE [12], Wernicke’s classic triad, autopsy evidence of WE, or clinical response to thiamine. The defining signs and symptoms for WE were dietary deficiencies, oculomotor abnormalities, cerebellar dysfunction, or an altered mental state. All case studies published in various levels of English were included in the systematic review.

### 2.2. Search Strategy and Study Selection

We searched MEDLINE, EMBASE, Scopus, and Web of Science using keyword terms (pediatric, child, infant, Wernicke Encephalopathy, and Korsakoff’s syndrome) from 1995 onward. All output was sorted by Zotero, and duplicates were removed. All titles and abstracts were reviewed through AI-based ordering principles via ASReview 1.5. The first author reviewed the title and abstracts of the search yield for eligibility and screened potentially eligible papers in full text to further assess eligibility. We also extracted data from eligible case studies in full text. See Figure 1.

We extracted and indexed the following data: year of publication, age, sex, etiology of PWE, presenting signs of WE, presence of ataxia in WE, presence of eye movement disorders (nystagmus and ophthalmoplegia) in WE, presence of mental status change in WE (described as confusion, memory deficits, altered consciousness or drowsiness, crying and irritability in infants), presence of the full triad of symptoms, hypotonia, loss of reflexes, hemiparesis, photophobia, hypotonia, hallucinations, spasms, incontinence, MRI/CT findings in WE, treatment for WE, and outcome. Cases were excluded if too little information was available to confirm a diagnosis of WE or no clinical characteristics regarding the patient or course of illness were available. We analyzed the data with SPSS (version 28.0). We calculated descriptive statistics (medians, ranges, SD, frequencies, and percentages) for article and patient demographics, prodromal signs and symptoms, clinical features of WE, treatment dosing, and cognitive outcome.

## 3. Results

### 3.1. Etiology of Pediatric Wernicke Encephalopathy

We extracted 88 pediatric case reports on Wernicke’s Encephalopathy from 80 published manuscripts [13,14,15,16,17,18,19,20,21,22,23,24,25,26,27,28,29,30,31,32,33,34,35,36,37,38,39,40,41,42,43,44,45,46,47,48,49,50,51,52,53,54,55,56,57,58,59,60,61,62,63,64,65,66,67,68,69,70,71,72,73,74,75,76,77,78,79,80,81,82,83,84,85,86,87,88,89,90,91,92]. In total, 23 case studies were published on PWE following cancer [13,14,15,16,17,18,19,20,21,22,23,24,25,26,27,28,29,30,31,32,33,34], moreover 19 case studies were published in gastrointestinal disease [8,35,36,37,38,39,40,41,42,43,44,45,46,47,48,49,50,51]. Seventeen cases suffered from malnutrition [52,53,54,55,56,57,58,59,60,61,62,63,64,65,66], twelve cases had psychiatric or neurodevelopmental disorders leading to restricted food intake [8,67,68,69,70,71,72,73,74,75,76,77], four cases developed WKS following obesity surgery [78,79,80,81,82,83], four had kidney disease [84,85,86,87], two had COVID-19 [88,89], one had severe complications in the PICU [92], one had hyperemesis gravidarum [91], and one had a genetic mutation as the primary consequence of pediatric WKS [92].

### 3.2. Prodromal Conditions of PWE

In Figure 2, the prodromal signs and symptoms prior to the development of PWE are represented as a percentage of the total number of case studies (n = 83). The most frequent prodromal symptoms were nausea (60%) and vomiting (55%). Of importance, 37% received parenteral nutrition formula without thiamine prior to the diagnosis of PWE. Other common symptoms prior to PWE were a loss of appetite (30%), abdominal pain (18%), and diarrhea (11%).

### 3.3. Signs and Symptoms of PWE

In Figure 2, the signs and symptoms of PWE are represented as a percentage of the total sample of case studies (n = 88). Clinical findings indicative for PWE include bilateral thalamic hyperintensities and/or hyperintensities in the mammillary bodies. Importantly, MRI was sensitive for PWE-selective findings in 85.2% of the cases. Of the classic triad, 79.6% showed eye movement disorders (66% had nystagmus and 50.3% ophthalmoplegia), 75% showed mental status change, and 63% had ataxia. The classic triad symptoms were absent in 3.4% of the cases. One symptom of the triad was present in 18.4% of the cases; 34.5% showed two signs of the triad, and 43.7% had a full triad. The most common additional symptoms were drowsiness (42.9%), altered reflexes (21.6%), hypotonia (19.3%), irritability of the child (18.2%), and tremors (13.6%).

Of the 88 cases, 35 cases were male and 48 cases were female (5 unknown sex). The age range was 0–18 years. Age was significantly correlated with the number of symptoms of the PWE triad (R(86) = 0.269, *p* < 0.05). Boys and girls had a comparable number of symptoms of the triad on average (t(81) = 0.043, *p* = 0.966).

### 3.4. Treatment and Outcome

Within the PWE cases, the course of treatment and outcomes were diverse. Of the 80 cases that described the course of treatment, 12.5% (10 cases) gave oral thiamine to treat PWE [17,19,25,26,34,37,52,56,58,91]. Of these 10 patients, 1 did not survive PWE [17], and 5 had residual motoric and mental symptoms following treatment [19,34,52,58,92], suggestive that oral thiamine prescription is not the adequate treatment for PWE.

More than half of the PWE patients (51.6%) received relatively low doses of parenteral thiamine (100 mg or less). Of importance, six case studies report on initial low parenteral doses (<100 mg/day) that were increased to higher doses (1000–1500 mg/day) and correction of magnesium deficits, leading to fast recovery after dosage increase [30,36,46,67,68,80]. Of the 85 cases reporting on outcome, 4.7% of the cases did not survive PWE, 34.1% of the patients had residual motoric or cognitive issues, and 61.2% were symptom-free after treatment.

## 4. Discussion

This systematic review highlights the diverse etiologies and clinical manifestations of pediatric Wernicke Encephalopathy (PWE), underlining the need for heightened awareness and prompt intervention in pediatric cases. Our review of 88 case studies revealed that while PWE can arise from various medical conditions, common etiologies include cancer, gastrointestinal diseases, malnutrition, and neurodevelopmental conditions, among others.

Wernicke Encephalopathy is a life-threatening neurological disorder caused by thiamine (vitamin B1) deficiency [4,7]. In the adult population, WE has traditionally been associated with chronic alcoholism [2,3,4]. In the pediatric population, cancer and gastrointestinal disease were most frequently reported on. Concerning the symptoms preceding PWE onset, nausea and vomiting emerged as the most frequent prodromal symptoms, present in 60.2% and 55.4% of cases, respectively. These symptoms, along with loss of appetite, abdominal pain, and diarrhea, suggest that gastrointestinal disturbances are critical early indicators of (severe) thiamine deficiency, in line with earlier research in adults [2,3,4,7].

The classic triad of WE—eye movement disorders, mental status changes, and ataxia—was present in varying degrees: 79.6%, 75%, and 63%, respectively. MRI demonstrated high sensitivity (85.2%) for PWE diagnosis, a notable difference from adult WE where MRI sensitivity is much lower. This suggests that MRI should be a standard diagnostic tool in suspected cases of PWE.

Of importance, 37.3% of the patients in this systematic review received parenteral nutrition without thiamine supplementation prior to PWE diagnosis, emphasizing the critical need for thiamine in such nutritional regimens to prevent WE. In many of the reports included in our review, vitamin B status was overlooked, and inadequate management of thiamine deficiency resulted in increased severity of the symptoms and ultimately PWE.

Treatment approaches varied, with a significant portion of cases (12.5%) initially treated with oral thiamine, leading to adverse outcomes in many. This finding underscores that oral thiamine is inadequate for PWE treatment. Higher parenteral doses of thiamine were often associated with faster and more complete recovery, reinforcing current guidelines recommending prompt high-dose intravenous treatment with thiamine repletion in WE cases [4,5,6]. Full remission was achieved in 61.2% of cases, with later publications showing better outcomes likely due to improved treatment protocols over time.

The findings of this review underscore the importance of early recognition of PWE prodromal symptoms, particularly gastrointestinal disturbances. Clinicians should ensure thiamine supplementation in at-risk pediatric populations receiving parenteral nutrition. Finally, parenteral thiamine administration should be the standard of care to optimize recovery and minimize long-term neurological sequelae.

## 5. Conclusions

Pediatric WE presents significant diagnostic and therapeutic challenges. Early recognition and appropriate treatment are crucial to prevent long-term adverse outcomes. Future research should focus on establishing standardized diagnostic and treatment protocols to further improve outcomes for pediatric patients with WE. Enhanced awareness and proactive management strategies are vital in improving outcomes and preventing long-term sequelae in affected patients.

## Figures and Tables

**Figure 1 pediatrrep-17-00015-f001:**
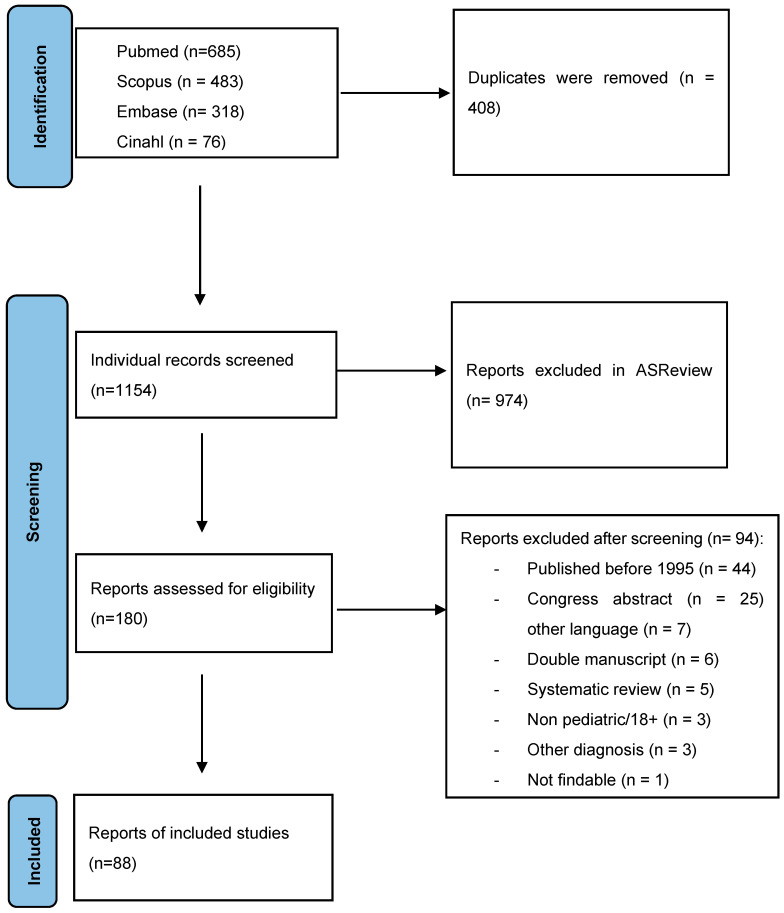
Flow-chart for inclusion of manuscripts on pediatric Wernicke Encephalopathy in this systematic review.

**Figure 2 pediatrrep-17-00015-f002:**
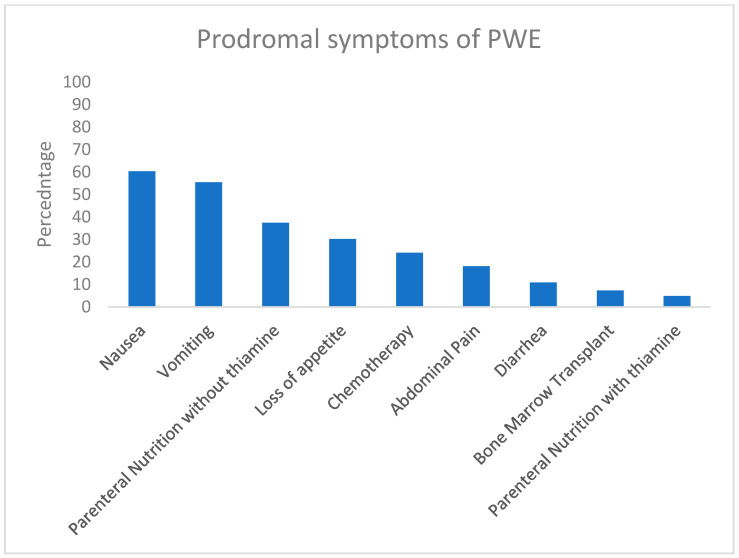
The prodromal symptoms leading to pediatric Wernicke Encephalopathy (PWE) in 88 case studies. The symptoms are presented as a percentage of all included cases.

## Data Availability

Data is contained in the article.

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
