# Peer review of "Pediatric Wernicke Encephalopathy: A Systematic Review"

_pediatrrep, 2025, doi:10.3390/pediatric17010015_

Round 1

Reviewer 1 Report

Comments and Suggestions for Authors

In the article titled "Pediatric Wernicke Encephalopathy: a systematic review" authors reviewed existing publications on PWE.

1. In the flow-chart figure authors should  show that 408 articles had duplicates which were removed.

There are several typos in the entire manuscript.

1. There are 2 figures in the manuscript and both of the figures were labeled as Figure 1.

2. In the methods section (Page 2; Line 88), ASReview is mistyped as ASRReview.

3. Page 5; Line 176-177 : hypotonia (19/3%)?  - Correct it to 19.3%.

4.  Page 6; Line211: MRI sensitivity is much lower 4. - Delete 4.

 Page 7 ; Line 245-246: In the authors contribution, it is mentioned that Jan W. Wijnia is responsible for funding acquisition.  Page 7; Line 248: Funding: There is no funding for this research project. If there is no funding for this manuscript, then please remove the author contribution statement for funding acquisition.

Author Response

  1. In the flow-chart figure authors should  show that 408 articles had duplicates which were removed.

Response: We have added this to figure 1.

There are several typos in the entire manuscript.

Response: We have corrected the manuscript for typos.

  1. There are 2 figures in the manuscript and both of the figures were labeled as Figure 1

    Response: We have labeled figure 2 as figure 2 in the revision.

2. In the methods section (Page 2; Line 88), ASReview is mistyped as ASRReview.

Response: we have corrected this typo.

3. Page 5; Line 176-177 : hypotonia (19/3%)?  - Correct it to 19.3%.

Response: we have corrected this typo

4.  Page 6; Line211: MRI sensitivity is much lower 4. - Delete 4.

Response: we have corrected this typo

 Page 7 ; Line 245-246: In the authors contribution, it is mentioned that Jan W. Wijnia is responsible for funding acquisition.  Page 7; Line 248: Funding: There is no funding for this research project. If there is no funding for this manuscript, then please remove the author contribution statement for funding acquisition.

Response:  We have removed the funding acquisition part.

Reviewer 2 Report

Comments and Suggestions for Authors

Dear Editor,

The submitted manuscript discuss about a review of Pediatric Wernicke Encephalopathy. After their initial screening procedure, they focused on 88 papers for their analysis. 

I have one main suggestion which can be included in the manuscript.

Please provide a table with sex (male or female) of the patients and a range of their age in these 88 different studies. Although their study is focused on the patients with less than 18 years of age, it is not clear whether there is any correlation between age and sex with the prodromal symptoms of PWE. They don't need to include all the symptoms. The authors can include only nausea, vomiting, parenteral nutrition without thiamine. This will help the readers to get a better understanding.

Thank you.

Author Response

The submitted manuscript discuss about a review of Pediatric Wernicke Encephalopathy. After their initial screening procedure, they focused on 88 papers for their analysis. 
I have one main suggestion which can be included in the manuscript. Please provide a table with sex (male or female) of the patients and a range of their age in these 88 different studies. Although their study is focused on the patients with less than 18 years of age, it is not clear whether there is any correlation between age and sex with the prodromal symptoms of PWE. They don't need to include all the symptoms. The authors can include only nausea, vomiting, parenteral nutrition without thiamine. This will help the readers to get a better understanding.

Response: We have included the following paragraph: Of the 88 cases, 35 cases were male and 48 cases female (5 unkown sex). The age range was 0-18 years. Age was significantly correlated with the number of symptoms of the PWE triad (R(86) = .269, p < .05). Boys and girls had a comparable number of symptoms of the triad on average (t(81) = .043, p = .966).

Reviewer 3 Report

Comments and Suggestions for Authors

Authors present a literature review on pediatric Wernicke encephalopathy in 88 patients with different ethiological background described in the literature in the past 30 years. MRI has shown high diagnostic sensitivity and classical symptoms were noted also in these patients: eye movement disorder, mental status change and ataxia. Main conclusion is that higher parenteral thiamine doses correlated with faster recovery and better outcomes,  Full remission was achieved in 61% of cases. This is a nice study which deals with a rare but potentialy fatal neurological disorder in children. Introduction is well written, study is registered in PROSPERO and search strategy is well explained. I suggest to include the Table with 88 cases with main characteristics of each patient, especially more discussion on MRI characteristics are warranted and I suggest to expand the discussion with this point. 

Author Response

I suggest to include the Table with 88 cases with main characteristics of each patient, especially more discussion on MRI characteristics are warranted and I suggest to expand the discussion with this point. 

Response: We have included additional information on the 88 participants in the text:  Of the 88 cases, 35 cases were male and 48 cases female (5 unkown sex). The age range was 0-18 years. Age was significantly correlated with the number of symptoms of the PWE triad (R(86) = .269, p < .05). Boys and girls had a comparable number of symptoms of the triad on average (t(81) = .043, p = .966).

Regarding MRI, we have included the classical finding in the text: Clinical findings indicative for PWE include bilateral thalamic hyperintensities, and/or hyperintensities in the mammillary bodies.